# Recovery and Fatigue Behavior of Forearm Muscles during a Repetitive Power Grip Gesture in Racing Motorcycle Riders

**DOI:** 10.3390/ijerph18157926

**Published:** 2021-07-27

**Authors:** Michel Marina, Priscila Torrado, Raul Bescós

**Affiliations:** 1Research Group in Physical Activity and Health (GRAFiS), Institut Nacional d’Educació Física de Catalunya (INEFC), Universitat de Barcelona (UB), 08038 Barcelona, Spain; priscilatorradopineda@gmail.com; 2Plymouth Institute of Health and Care Research (PIHR), University of Plymouth, Plymouth PL4 8AA, UK; raulbescos@gmail.com

**Keywords:** handgrip, carpi radialis, flexor digitorum superficialis, neuromuscular fatigue, motorcycle, recovery

## Abstract

Despite a reduction in the maximal voluntary isometric contraction (MVC_isom_) observed systematically in intermittent fatigue protocols (IFP), decrements of the median frequency, assessed by surface electromyography (sEMG), has not been consistently verified. This study aimed to determine whether recovery periods of 60 s were too long to induce a reduction in the normalized median frequency (MF_EMG_) of the flexor digitorum superficialis and carpi radialis muscles. Twenty-one road racing motorcycle riders performed an IFP that simulated the posture and braking gesture on a motorcycle. The MVC_isom_ was reduced by 53% (*p* < 0.001). A positive and significant relationship (*p* < 0.005) was found between MF_EMG_ and duration of the fatiguing task when 5 s contractions at 30% MVC_isom_ were interspersed by 5 s recovery in both muscles. In contrast, no relationship was found (*p* > 0.133) when 10 s contractions at 50% MVC were interspersed by 1 min recovery. Comparative analysis of variance (ANOVA) confirmed a decrement of MF_EMG_ in the IFP at 30% MVC_isom_ including short recovery periods with a duty cycle of 100% (5 s/5 s = 1), whereas no differences were observed in the IFP at 50% MVC_isom_ and longer recovery periods, with a duty cycle of 16%. These findings show that recovery periods during IFP are more relevant than the intensity of MVC_isom_. Thus, we recommend the use of short recovery periods between 5 and 10 s after submaximal muscle contractions for specific forearm muscle training and testing purposes in motorcycle riders.

## 1. Introduction

Simulation of highly repetitive intermittent muscle contractions present during motorcycle competitions is currently under investigation because of their relationship with the development of clinically significant conditions, especially in the hand/forearm. These conditions, characterized by pain and loss of the hand or forearm function, are defined as exertional compartment syndrome [1,2,3,4]. They frequently lead to long periods of illness in motorcycle riders, especially in those participating in endurance competitions such as 24 h races, where they must brake more than 4000 times and make 10,000 gear changes [5]. Similar pathological patterns can also occur among workers in the manufacturing industry [6]. The fact that many athletes, manual workers, and musicians must endure their mechanical work over long periods of time, muscle contraction intensities that characterize each activity explains the large number of studies focused on neurophysiological fatigue of the forearm muscles [7,8,9,10,11]. These muscles are involved in a great variety of repetitive grip tasks that can lead to neuromuscular fatigue and functional impairment when these tasks become chronic. Thus, it is important to obtain better knowledge and understanding of the mechanisms involved in these physiological situations to prevent forearm syndrome.

When assessing human muscle fatigue with superficial electromyography (sEMG), the power spectrum displacement towards lower frequencies has been extensively documented in continuous fatiguing protocols (CFP), in which submaximal voluntary contractions are maintained until exhaustion [9,12,13,14,15,16]. Intermittent fatiguing protocols (IFP) have also been extensively studied because intermittent contractions at different intensities are very common in the everyday life of the majority of workers and athletes [17,18]. Consequently, when comparing both types of fatiguing tasks (CFP versus IFP) specifically adapted to motorcycle riders [9], IFP showed a stronger relationship with the level of motorcyclist forearm discomfort compared to CFP [9].

The relative intensity of the contraction with respect to maximal voluntary isometric contraction (MVC_isom_) registered in a non-fatigued condition (%MVC_isom_) is a key factor that modulates muscle fatigue. Studies looking at CFP confirmed that the higher the intensity of the effort, the shorter the time to task failure [14,16,19], obviously because of the lack of recovery periods. Moreover, it has been generally observed that %MVC_isom_ and the time to task failure (also called time limit) have a significant effect on the decrement of sEMG frequency (MF_EMG_) and increment of the sEMG amplitude (RMS_EMG_) [14,15,20]. A reduction in MF_EMG_ was observed during CFPs at 10% MVC [21,22], 25% MVC [22,23,24], 30% [21], 40% [9,21,24], 60% [25], 55, 70, 80, and 90% of MVC [24]. Nevertheless, some caution is recommended in regard to MF_EMG_ because %MVC_isom_ should not be considered as a definitive factor explaining the absence of a reduction in MF_EMG_ during fatiguing protocols [24,25].

A second factor that is necessary to consider when measuring fatigue is the duration of the effort or exertion time. It is known that the duration of fatiguing tasks at a constant relative submaximal %MVC_isom_ is negatively associated with MVC_isom_ decrements, reaching the maximal point at time to task failure [26]. Duration of the effort induces a linear decrement of MF_EMG_ [14,24,27] whose slope may differ slightly depending on the muscle group and type of movement [15,16,20,21]. With %MVC_isom_ and duration of the effort as the main triggers of fatigue in CFPs, the greatest MF_EMG_ decrements were observed at longer durations due to lower %MVCs [28].

A third factor must be taken into account in IFP: the duration of the recovery interspersed between muscle contractions. Controversial MF_EMG_ results have been observed when applying IFPs, despite the lower MVC_isom_ recorded at the end of such fatigue protocols. For example, some authors, but not all [29,30], reported a reduction in MF_EMG_ during an IFP [31,32]. MF_EMG_ was similar to pre-fatigue values with different work–rest cycles, whatever the intensity used in the IFP, [22]. These results are consistent with the findings of Mundale [28], who also studied the factors that lengthen the endurance time of an IFP. It seems that the duration of the recovery period could be one of the key factors explaining the disparity in MF_EMG_ results, particularly among IFPs. Looking at motorcycle riders, we [5] observed no significant MF_EMG_ decrement throughout a 24-h motorcycle endurance race despite the significant decrease in MVC_isom_. Following the recommendation of previous studies [33,34,35], we took care not to exceed an interval of 4–5 min between the end of each relay and the handgrip assessment. The lack of MF_EMG_ decrement led to conclude that this interval was too long. According to these findings, we decided to compare an IFP and CFP specifically adapted to motorcycle riders [9]. The lack of a reduction in MF_EMG_ in the IFP suggested that rest cycles were too long, achieving basal values of MF_EMG_ between the work cycles. These findings are in agreement with another study by Krogh-Lund and Jorgensen [23] that compared two pairs of fatiguing sustained isometric contractions at 40% MVC_isom_ separated by different rest intervals. They found that the MF_EMG_ at the start of the second contraction did not recover to pre-fatigued values when the rest interval was less than 1 min, [23]. Other studies reached similar conclusions when they used intermittent contractions [24,25,36], suggesting that a MF_EMG_ shift toward the pre-fatigue state occurs independently of the contraction intensity (25–50%) [36].

Some authors [37] suggest that the validity of the spectral shift of the sEMG signal in assessments of fatigue must be taken with caution because a clear MVC decrement is sometimes weakly reflected in the sEMG signal [38]. This is supported by studies that used IFP to assess muscle fatigue [29,30,39,40]. In contrast, the usefulness of the sEMG signal for studying muscle fatigue in occupational field studies [41] is supported by other studies that reported a reduction in MF_EMG_ with IFP [31,32]. These overall discrepancies between studies suggest that the combination of different, contraction–relaxation periods, effort intensities (%MVC_isom_), muscle groups, and other non-controlled or non-reported factors, are critical to understanding muscle fatigue in IFPs [18,22,42].

Therefore, this study aimed to verify in road racing motorcycle riders whether the recovery period performing an IFP matching the braking movement was more relevant than the contraction intensity and effort duration in two forearm muscles (flexor digitorum superficialis and carpi radialis). We hypothesized that MF_EMG_ will not decrease during the contractions performed at 50% MVC_isom_ because they are preceded by long recovery periods. On the contrary, MF_EMG_ recorded at 30% MVC_isom_ and during a shorter exertion time (5 s) may decrease due to short recovery periods (5 s).

## 2. Methods

### 2.1. Subjects

Twenty-one road racing motorcycle riders aged 29.1 ± 8.0 years (body mass: 72.1 ± 5.5 kg; height: 176.2 ± 4.9 cm) participated in this study. Of these riders, 48% were winners within the Spanish and/or World Championships and 24% were on the podium of the Championship at the end of the season over the previous 6 years. The remaining 28% participated in races at the regional level with at least 5 years of racing experience. The study was approved by the Clinical Research of the Ethics Committee for Clinical Sport Research of Catalonia (Ref. number 15/2018/CEICEGC) and written consent was given by all the participants. The data were analyzed anonymously, and the clinical investigation followed the principles of the Declaration of Helsinki.

### 2.2. Procedures

Before the assessment, the brake lever to handgrip distance was adjusted to the participant’s hand size to ensure that hand placement in relation to the brake was similar across all subjects. Afterwards, during the familiarization period, the subject practiced six to ten submaximal non-stationary contractions while watching the dynamometric feedback displayed on the PC screen, while the researcher provided feedback about how to interpret the auditory and visual information. A continuous linear feedback and a columnar and numerical display showed the subject the magnitude of the force they exerted against the brake lever. In addition, a different tone was provided depending on the force level. Dynamometric and sEMG signals were recorded and these signals were synchronized with an external trigger. Five minutes before the beginning of the intermittent fatigue protocol (IFP), two MVC_isom_ trials separated by a 1-min rest were performed to provide a baseline value of MVC_isom_. The 1-min resting period between the two MVC_isom_s was considered sufficient to avoid fatigue from the previous contraction [43,44]. The higher MVC_isom_ was recorded as the basal value of that day and used to calculate the submaximal efforts (50% and 30% of the maximum). During the IFP, the subject adopted the “rider position” with both hands on the handlebar.

### 2.3. Sequence and Structure of the IFP

The intermittent protocol comprised a succession of a maximum of 25 rounds. Each round comprised two sections (Figure 1A). Section one consisted of six 5-s voluntary contractions of 30% MVC_isom_, with a resting period of 5 s between each contraction. Section two comprised a 3-s MVC_isom_ followed by a 1-min resting period and a 50% MVC_isom_ maintained for 10 s. During the 1-min resting period subjects were in the seated position with their hands resting on their thighs.

Intensities ranging from 10% to 40% of MVC have previously been used to carry out a continuous or intermittent fatigue protocol [18,22,45]. A sequence of 30% of MVC_isom_ was finally adopted after consulting with expert riders (exclusively, winners of races at the national and world level) who agreed about the perception of applying approximately this percentage of force during very strong braking in real situations.

Section two was designed to replicate an experimental protocol from one of our previous studies of motorcycle riders [5]. The test stopped when the subject was unable to maintain the established 50% of MVC_isom_ for 10 s, or the concurrent MVC_isom_ was 10% lower than 50% of the MVC_isom_ value. The number of rounds achieved by each subject was used as a performance measure.

### 2.4. Dynamometric Assessment

To simulate the overall position of a rider on a 600–1000-cc racing motorcycle, a static structure was built to preserve the distances between the seat, stirrups, and particularly the combined system of shanks, forks, handlebar, brake and clutch levers, and gas (Figure 2). As it happens in a road race motorcycle, levers tilt, distances between levers and handle gas, and distance between the handlebar and seat were modified according to the ergonomic requirements of the rider (Figure 2).

The subjects were asked to exert a force against the brake lever (always the right hand) using the second and third finger to hold the lever half way, and the thumb and other fingers grasping the handgrip at the same time, which is the most common way of braking of road racing motorcycle riders (Figure 2). Both arms had a slight elbow flexion (angle 150–160°), forearms half-pronated, wrist in neutral abduction/adduction position and alienated with respect to the forearm, dorsal flexion of the wrist no bigger than 10°, and legs flexed with feet above the footrests; in short, the typical overall position of a rider piloting a motorcycle in a straight line.

Special attention was given to controlling the handgrip position, and the wrist, elbow, and trunk angles to avoid any modification of the initial overall body position during the test. One experimenter supervised the recording of force and sEMG signals, and another continuously checked the maintenance of body position. It has been reported that variations in body posture [46] and wrist angles [47] alter the behavior of the forearm muscles during handgrip force generation.

To measure the force exerted against the brake lever we used a unidirectional gauge connected to the MuscleLab^TM^ system 4000e (Ergotest Innovation AS, Stathelle, Norway). The frequency of measurement was 400 Hz, and the loading range was from 0 to 4000 N. The gauge (Ergotest Innovation AS, Norway), with a linearity and hysteresis of 0.2%, and 0.1 N sensibility, was attached to the free end of the brake lever in such a way that the brake lever system and the gauge system laid over the same plane and formed a 90° angle approximately when the subject was exerting force. The MVC at the end of the IFP was compared to the MVC in the pre-fatigued state. The 30% and 50% MVC contractions were used for sEMG analysis.

### 2.5. Electromyography

A ME6000 electromyography system (Mega Electronics, Kuopio, Finland) was used to register flexor digitorum superficialis (FS) and carpi radialis (CR) EMG signals. Adhesive surface electrodes (Ambu Blue Sensor, M-00-S, Ballerup, Denmark) were placed 2 cm apart (from center to center) according to the anatomical recommendations of the SENIAM Project [48,49]. The raw signal was recorded at a sampling frequency of 1000 Hz. Data were amplified with a gain of 1000 using an analog differential amplifier and a common-mode rejection ratio of 110 dB. The input impedance was 10 GΩ. A Butterworth bandpass filter of 8–500 Hz (–3 dB points) was used. To compute the median frequency (MF_EMG_, Hz), Fast Fourier Transform was used with a frame width at 1024, a shift method of 30% of the frame width, and the “flat-topped” windowing function. The power spectrum densities were computed and averaged afterwards to obtain one mean or median for each submaximal contraction of 30% MVC_isom_ (5 s duration) and 50% MVC_isom_ (10 s duration). Afterwards, the median frequency (MF_EMG_) was normalized with respect to the basal condition during the MVC_isom_.

In order to obtain the same number of MF_EMG_ values from the IFP of each individual, and for each round and MVC_isom_ intensity, the six 30% MVC_isom_s of the first section (Figure 1A) were averaged to obtain one MF_EMG_ (MF_EMG30_). Each MF_EMG30_ was paired with the only MF_EMG_ of the second section (Figure 1A) obtained from the 50% MVC_isom_ (MF_EMG50_).

### 2.6. Statistics

Parametric statistics were used after confirming the normal distribution of the normalized parameters used in this study (MVC_isom_, MF_EMG30_, and MF_EMG50_) with the Shapiro-Wilk test. Descriptive results were reported as the mean and standard deviation. A paired sample t-test was used to compare the MVC_isom_ in the pre-fatigued state and at the end of the IFP. Two methodological approaches were used to verify the study’s hypothesis. First, we used regression analysis for each individual, to study the strength of the relation and detect possible trends between the number of rounds accomplished (independent variable) and the MF_EMG30_ (dependent variable). Second, we used a 2 (time points: T_1_ and T_2_) × 2 (muscles: FS and CR) × 2 (%MVC_isom_: 30 and 50) ANOVA of repeated measures to compare all MF_EMG_ values at the beginning and the end of the IFP, and to study potential interactions with the two muscle groups analyzed (CR and FS) and the two intensities that were preceded by distinct recovery periods (5 s for 30% MVC_isom_ and 1 min for 50% MVC_isom_). When necessary, the Greenhouse-Geisser’s correction was used if the sphericity test to study matrix proportionality of the dependent variable was significant (*p* < 0.05). Then, when a significant effect was found, a post-hoc analysis was carried out conducting multiple comparisons between the normalized rounds with Sidak’s adjustment. Partial Eta squared (η^2^p) was used to report effect sizes (0.01 ≈ small, 0.06 ≈ medium, >0.14 ≈ large). Statistical analysis was performed using the PASW Statistics for Windows, Version 18.0 (SPSS, Inc., Chicago, IL, USA). The level of significance was set at 0.05.

## 3. Results

At baseline conditions, MVC_isom_ (276 ± 46.6) was 53% lower than the MVC_isom_ at the end of the IFP (147 ± 46.3; *p* < 0.001).

Individual regression analysis (Table 1, Figure 3) was conducted to verify possible trends between the NMF of the CR and FS and the number of rounds accomplished by the motorcycle riders during an intermittent fatigue protocol (IFP) at two different intensities (30% and 50% of MVC_isom_). The overall individual regression analysis showed a significant linear relationship (*p* < 0.005) between the MF_EMG_ and the number of rounds accomplished by both muscles when they were exercised at 30% MVC_isom_ (CR_30_ and FS_30_), with pauses of 5 s between each contraction. In contrast, when both muscles were exerted at 50% MVC_isom_ (CR_50_ and FS_50_), after 1 min of recovery, no significant relationship was observed (*p* > 0.133). The higher correlation observed in CR_30_ and FS_30_ (*r* ≥ −0.71) in comparison to CR_50_ and FS_50_ (*r* ≤ 0.59) supports the hypothesis of a weaker relationship between the MF_EMG__50_ and the number of rounds when both muscles had the opportunity to recover for longer (1 min for CR_50_ and FS_50_). Similarly, the overall individual regression analysis showed that the fraction of MF_EMG_ variance, explained by the number of rounds attained during the intermittent protocol, was bigger with CR_30_ and FS_30_ (*r*^2^ ≥ 0.50) in comparison to CR_50_ and FS_50_ (*r*^2^ ≤ 0.40) (Table 1).

In addition to the regression analysis performed for each individual, Table 2 reveals that a greater number of riders satisfied better levels of statistical condition in CR_30_ and FS_30_ in comparison to CR_50_ and FS_50_. Moreover, the higher correlation values (*r* > 0.70) and higher levels of significance (*p <* 0.001) were associated with higher frequency values in CR_30_ and FS_30_, while lower correlation values (*r* < 0.39) and lower levels of significance (*p* > 0.05) were associated with a higher number of riders in CR_50_ and FS_50_.

Figure 3 is an example of the regression analysis carried out in one subject showing higher MF_EMG_ values for the CR in comparison to the FS. Moreover, at 50% MVC, the MF_EMG_ of the CR never dropped below the MF_EMG_ level established during the basal assessment (Figure 3B), which is consistent with the comparative results (Table 3).

The second methodological approach was used to determine whether less intense and shorter muscle contractions (30% MVC_isom_ instead of 50%; 5 s instead of 10 s) could induce bigger MF_EMG_ decrements in the CR and FS. The second objective was to determine whether the two muscles (CR and FS) had a similar MF_EMG_ decrement due to fatigue. Thus, we compared two times of measurement (T_1_ and T_2_), two muscles (CR and FS) and two contraction intensities (30% and 50% of MVC_isom_) (Table 3).

A significant three-way interaction was found (*p* < 0.001) with a large effect size (η^2^p = 0.5) (Table 3). Paired comparisons found lower values for the FS than the CR at both times and both intensities. Moreover, we observed a higher MF_EMG_ in the CR muscle at 30% MVC_isom_ (CR_30_) than at 50% MVC_isom_ (CR_50_) at the beginning of the IFP, but the opposite response was observed at the end. Finally, regarding the CR, while MF_EMG_ was lower at the end than at the beginning of the IFP at the 30% MVC_isom_ (CR_30_), the opposite was observed at the 50% MVC_isom_ exertion (CR_50_) (Table 3).

In addition, a significant two-way interaction was found between the time and MVC_isom_ intensity (time per intensity) with a large effect size (η^2^p = 0.63), but not for the other interactions (time per muscle, and intensity per muscle) with a small and medium effect size, respectively (Table 3). The MF_EMG_ was higher at the beginning than at the end of the IFP when both muscles were exerted at 30% MVC_isom_, but no significant differences were observed when they were exerted at 50% MVC_isom_. Finally, we observed a significant main effect for intensity and muscle factor (Table 3).

## 4. Discussion

The MVC_isom_ decrement observed in our IFP confirmed the occurrence of muscle fatigue as this physiological phenomenon is commonly defined as the “loss of the maximal force-generating capacity” [37,50]. From a functional and neurophysiological point of view, and according to the literature, the decrement of the sEMG power spectrum is related, among other factors, to: (1) a reduction in the conduction velocity of the active fibers [35]; (2) impairment of the excitation–contraction coupling [27] related to metabolic changes that occur during fatigue [51]; (3) the recruitment of new units [52], based on the knowledge that subjects with a high relative number of fast twitch fibers may have higher sEMG frequency values [53], and that during fatigue, they show a greater shift towards lower MF_EMG_ compared to subjects with a low relative number of fast twitch fibers [54]; (4) structural damage to muscle cells when muscle soreness is reported by the subjects [18]; (5) other reactions taking place beyond the muscle cell membrane [55], based on observations that short resting periods between each muscle activation are sufficient to maintain the neuromuscular excitability at normal levels during IFP. It must be highlighted that this study did not intend to explain the changes in MF_EMG_ induced by fatigue from a physiological perspective, we were focused on the relationship between the MF_EMG_ and the two factors controlled in our IFP: the load intensity and the work–rest cycle.

High variability of MF_EMG_ values at low loads has been attributed to the influence of the number of recruited muscle fibers and the synchronism and firing rate [56]. According to this, it could be more difficult to find a significant pattern at 30% MVC_isom_ rather than 50% MVC_isom_, but we found that the MF_EMG_ of the CR and FS decreased more consistently throughout the IFP when the muscles were exerted at 30% MVC_isom_ in comparison to 50% MVC_isom_. The regression analysis of each individual revealed systematically stronger correlations, coefficients of determination, and statistical significance with CR_30_ and FS_30_ in comparison with CR_50_ and FS_50_. Moreover, participants reported a stronger relationship between the number of rounds accomplished and the MF_EMG_ at 30% MVC_isom_, rather than 50% MVC_isom_, in both muscles that were assessed. In agreement with this, we found a higher and more significant MF_EMG_ decrement when the participants performed the IFP at 30% MVC_isom_, which may suggest different neuromuscular fatigue patterns between the CR_50_ and FS_50_ during the IFP [9]. If force intensity was the only one factor explaining these differences, it would be difficult to argue that time to exhaustion of any fatigue protocol would be longer when muscles work at higher intensities. As expected, other studies proved the opposite [22,23,57]. Moreover, when studying the magnitude of fatigue in two different IFPs at two different intensities (25 and 50% MVC_isom_), Seghers and Spaepen [42] observed very similar relative MF_EMG_ decrements in the two muscles analyzed (IFP at 25% MVC_isom_: 29%, and 30%; IFP at 50% MVC_isom_: 29%, and 28%), when sustaining an isometric contraction at 75% of prefatigued MVC_isom_ at the end of both protocols [42]. On the other hand, whereas the same authors observed a significant negative slope of the MF_EMG_ during the IFP at 25% MVC_isom_, during the IFP at 50% MVC_isom_ the slope did not differ significantly from zero. It is possible that the differences in MF_EMG_ changes during the two IFPs could be more related to differences in their work–rest cycles (10 + 10 s in 25% MVC_isom_ and 5 + 15 s in 50% MVC_isom_) than in the contraction intensity. In rock climbers, the significant reduction in the MF_EMG_ observed during an intense IFP (80% MVC_isom_) [58], with a work–rest cycle of 5 + 5 s (same cycle as in our IFP for the 30% MVC_isom_), indicates that the majority of the frequency components of the MF_EMG_ are unaffected by tension [24]. Thus, we believe that the key point for understanding the different MF_EMG_ patterns during our IFP must be the resting period before the two intensities. Only 5 s of recovery were interspersed between braking muscle contractions of the forearm at 30% MVC_isom_ compared to the 60 s (1 min) at 50% MVC_isom._ This clearly indicates that MF_EMG_ can be explained to a greater extent when the riders have a very short recovery time despite a smaller contraction intensity (30% MVC_isom_ instead of 50% MVC_isom_) and a shorter contraction time (5 s for 30% MVC_isom_ instead of 10 s for 50% MVC_isom_). Similar results were reported by Nagata et al. [25]. 

Nevertheless, it is important to highlight that these authors used a continuous fatigue protocol in which the force was maintained at an intensity of 60% MVC_isom_ until exhaustion, which substantially differs to the IFP in our study.

Before undertaking this study, it was not evident that 1 min of recovery before the 50% MVC_isom_ could be long enough to allow a systematic recovery of the MF_EMG_ towards baseline levels (pre-fatigued). The MF_EMG_ recovery curve towards pre-fatigued values can be characterized by an exponential function [59,60,61], as well as a logarithmic course characterized by large inter-individual variations [61,62]. Therefore, a large proportion of the MF_EMG_ spectrum recovery corresponds to the first 1 min of the exponential recovery curve [21,23,43,44,59,60,61,62,63]. However, depending on the fatigue protocol, this does not mean full restoration comparable to pre-fatigued or basal MF_EMG_ values. Following the completion of ten cycles of work/rest (10 s/10 s) at MVC_isom_, Mills [59] observed that the mean power frequency of a compound muscle action potential evoked by supramaximal nerve stimulation required 3 min to recover 50% of its initial values. Three to six minutes, depending on age, are sometimes necessary to recover the pre-fatigued MF_EMG_ values of the abductor digiti-minimi muscle after a MVC_isom_ exertion maintained until 50% MVC_isom_ [64]. Other studies [62,65,66] have confirmed that the majority of the MF_EMG_ spectrum is re-established after 1 and 3 min of recovery, but full recovery it may take until the fifth minute [23,62]. Interestingly, Krogh-Lund and Jorgensen [23] observed that the restoration of MF_EMG_ paralleled that of conduction velocity for the last 4 min of recovery. Regarding the first part of the exponential recovery curve, 35 s were sufficient to allow restoration of 50% of the decline in MF_EMG_ during the previous fatigue protocol [61], but a longer interval (1.4 min) was required to reach 50% of pre-fatigued values for the biceps brachii [67]. Faster MF_EMG_ recovery (up to 85% of the pre-fatigued state during the first minute) was found by Krogh-Lund [21] in the brachioradialis and biceps brachii muscles. Nevertheless, the standard error of the measurement (about 60 s) reported by Elfving et al. [61], which was much larger than the average recovery, reflects the large between-subject variability of the MF_EMG_ parameter when studying the recovery phase. The inconclusive results reported in the literature combined with the accepted large variability that characterizes this type of analysis, support the idea that different combinations of IFP (contraction intensities and durations of contraction and relaxation) to assess muscle fatigue can provide different results [42]. Thus, although it is difficult to compare sEMG data from different studies it is even more complicated when the protocol involves voluntary exercise [37]. The fact that the physiological mechanisms causing muscle fatigue are specific to the task [68], should encourage future studies looking at road racing motorcycle riders to focus on the specific conditions of the forearm muscles, in order to understand better pathologies such as exercise-induced compartment syndrome.

The main limitations of this study were that effort duration, contraction intensity, and recovery time were not separated in different IFPs. Ideally, swapping these three factors would mean that riders had to attend the laboratory on at least six occasions to undertake different IFPs and following a randomized protocol. However, this approach was not feasible in the current study due to the busy racing and training schedules and other commitments of the population of this study.

## 5. Conclusions

This study reproduced, in the most accurate way and under laboratory conditions, the braking action in road racing motorcycle riders to investigate different work–rest cycles during an IFP. For training purposes, we recommend using short recovery periods between 5 and 10 s after submaximal muscle contractions as the most effective way to induce muscle fatigue than intermittent tasks performed at higher intensities and with longer recovery periods. That is, much less than 1 min for the resting time (no more than 30 s) according to the results of previous studies [21,23,43,44,60,61,63]. Furthermore, contraction intensities above 50% MVC_isom_ may not be useful for road racing motorcycle riders since only around 30% MVC_isom_ is required to break in real conditions when they have to slow down at high speed (more than 270 km/h) to connect a straight line with a slow curve [9]. Muscle contraction times longer than 10 s are not useful either to match road racing requirements, so protocols involving this type of contraction are not recommended for these individuals. Finally, accelerations with the right hand promote hand dorsal flexion and the assessment of both movements (braking and acceleration) have not been combined in a single IFP. This must be taken into account in future studies to match the real conditions of road motorcycle racing in laboratory settings. This knowledge is needed to enhance our understanding of the most appropriate stimulus (muscle contraction intensities and recovery periods) to be applied within the training programs of road racing motorcycle riders in order to mimic racing conditions and to reduce the risk of muscle pathologies such as the forearm chronic exertional compartmental syndrome.

## Figures and Tables

**Figure 1 ijerph-18-07926-f001:**
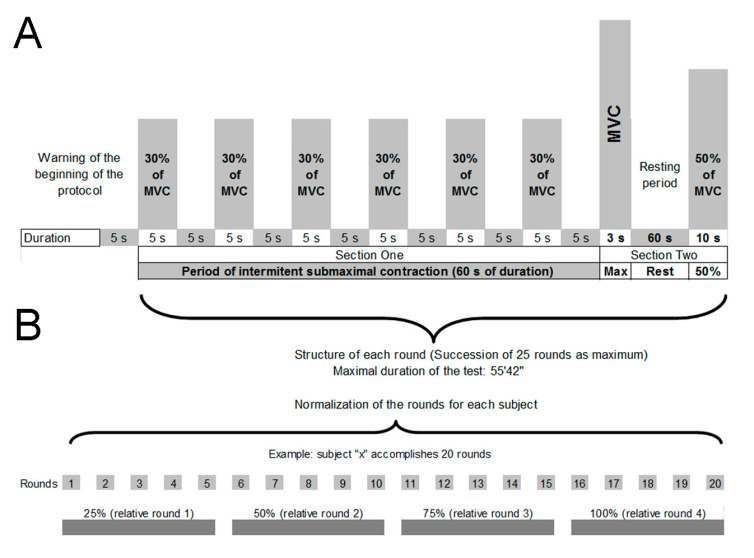
(**A**): Description of the sequence and structure of the intermittent protocol. Auditory feedback was provided to ensure the exact duration of each contraction and resting period. (**B**): Represents an illustration of a subject who performed 20 rounds, which means that each one of the four successive relative rounds is composed of five rounds.

**Figure 2 ijerph-18-07926-f002:**
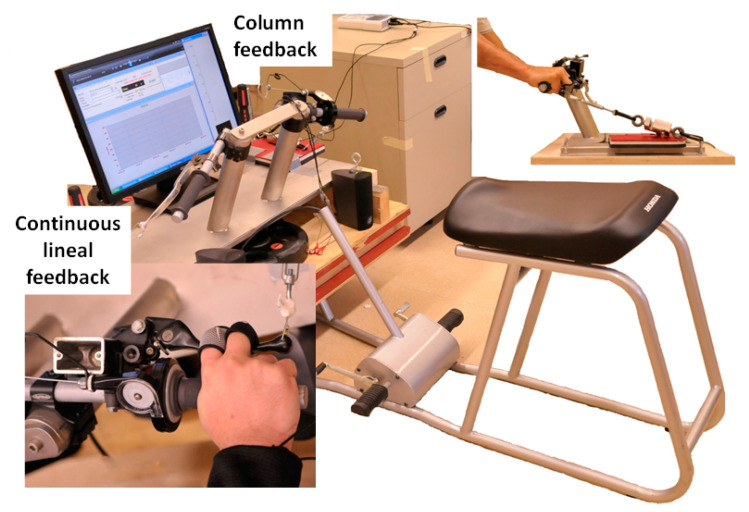
Simulation of the overall position of a rider above a motorcycle race from 600 cc to 1000 cc. A static structure was built to preserve the distances between seat, stirrups, and particularly the combined system of shanks, forks, handlebar, brake and clutch levers, and gas.

**Figure 3 ijerph-18-07926-f003:**
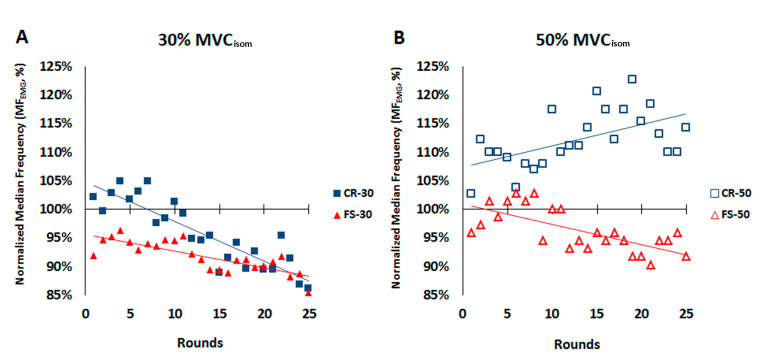
Example of a comparative regression analysis of an individual. Regression of the carpi radialis (CR) and flexor superficialis digitorum (FS) at the two intensities: (**A**) is 30% of MVC_isom_, and (**B**) is 50% of MVC_isom_; both used in the intermittent protocol.

**Table 1 ijerph-18-07926-t001:** Regression analysis of normalized median frequency (MF_EMG_, dependent variable), against the number of rounds (independent variable) accomplished by each rider (*n* = 21). Muscles analyzed are the carpi radialis (CR) and flexor digitorum superficialis (FS) at 30% and 50% of MVC.

*n* = 21		*r*	*r^2^*	Error of Estimate	*F*	*p*
CR_30_	Mean sd	−0.756 ± 0.176	0.580 ± 0.266	0.026 ± 0.012	54.163 ± 57.827	0.005 ± 0.009
	CI _sup_ CI _inf_	0.758 0.753	0.583 0.576	0.027 0.026	54.954 53.372	0.006 0.005
CR_50_	Mean sd	0.594 ± 0.284	0.397 ± 0.302	0.045 ± 0.019	28.046 ± 43.913	0.133 ± 0.295
	CI _sup_ CI _inf_	0.598 0.590	0.401 0.393	0.045 0.045	28.647 27.445	0.137 0.129
FS_30_	Mean sd	−0.711 ± 0.152	0.504 ± 0.214	0.022 ± 0.008	27.659 ± 23.267	0.005 ± 0.007
	CI _sup_ CI _inf_	0.713 0.709	0.507 0.501	0.022 0.002	27.977 27.341	0.005 0.004
FS_50_	Mean sd	−0.542 ± 0.283	0.338 ± 0.290	0.033 ± 0.016	20.524 ± 31.906	0.158 ± 0.288
	CI _sup_ CI _inf_	0.546 0.539	0.342 0.334	0.033 0.033	20.960 20.087	0.161 0.154

Pearson coefficient correlation (*r*), R squared *(r*^2^), error of the estimate, F-statistics (*F*), level of significance (*p*), degree of freedom (df: 1, 10–23). The minor number of accomplished rounds was 10. Five riders succeeded to perform all 25 rounds of the intermittent protocol.

**Table 2 ijerph-18-07926-t002:** Frequency table. Number of motorcycle riders who match the condition reported in the individual linear regression analysis. Normalized median frequency (MF_EMG_) was the variable taken for analysis against the number of rounds accomplished during the intermittent fatigue protocol.

*n* = 21	*r*			*p*		
	>0.70	0.40–0.69	<0.39	<0.001	0.001–0.05	ns
CR_30_	13	8	0	14	7	0
CR_50_	10	7	4	10	7	4
FS_30_	13	8	0	12	9	0
FS_50_	7	7	7	9	5	7

**Table 3 ijerph-18-07926-t003:** 2 (Time) × 2 (Muscles) × 2 (% MVC_isom_) ANOVA of repeated measures between the beginning (T_1_) and the end (T_2_) of the intermittent fatiguing protocol (IFP). The parameter of analysis is the normalized median frequency (MF_EMG_) of the Carpi Radialis (CR) and Flexor Digitorum Superficialis (FS).

Effect	*F*	*df*	*p*	η^2^p	Paired Comparisons	*p*
T × In × M	20.04	1, 20	<0.001	0.5	T_1_ & T_2_: FS_30_ < CR_30_; FS_50_ < CR_50_	<0.001
					T_1_: CR_30_ > CR_50_; T_2_: CR_30_ < CR_50_	<0.002
					CR_30_: T_1_ > T_2_; CR_50_: T_1_ < T_2_	<0.001
T × In	33.6	1, 20	<0.001	0.63	In_30_: T_1_ > T_2_	<0.001
					In_50_: T_1_ < T_2_	<0.024
T × M	0.74	1, 20	ns	0.04		
In × M	3.02	1, 20	ns	0.13		
T	1.43	1, 20	ns	0.07		
In	28.58	1, 20	<0.001	0.59		
M	42.43	1, 20	<0.001	0.68		

Time (T), Intensity (In) of 30% MVC_isom_ (In_30_) and 50% MVC_isom_ (In_50_), Muscle (M).

## Data Availability

The data presented in this study is not available.

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
