# Peer review of "Recovery and Fatigue Behavior of Forearm Muscles during a Repetitive Power Grip Gesture in Racing Motorcycle Riders"

_ijerph, 2021, doi:10.3390/ijerph18157926_

Round 1

Reviewer 1 Report

The authors make an interesting and well-designed and realized proposal. My congratulations. Minor comments have been described in the attached document.

Author Response

  • I recommend indicating in the title the chosen population (ie. Racing motorcycle riders). We appreciate the reviewer’s comment and this has been done.
  • I suggest changing the term MVC to MVIC (add the isometric concept). We appreciate this suggestion. We agree it is important differentiating isometric from concentric or eccentric contraction. We have replaced MVC by MVCisom into the revised article.
  • This information I suggest synthesizing it or transferring it to discussion (lines 76-92). We appreciate the reviewer’s comment which is in line to another comment of reviewer 2. We have made an effort to strengthen the structure of the introduction following this order:
    1. Justification of the study and association of repetitive handgrip action with functional forearm disorders (lines 24-37).
    2. Utility of focusing on IFPS because they are “common in the every-day life activities” (lines 41-43).
    3. Discrepancies between continuous and intermittent fatiguing protocols (CFP & IFP respectively; lines 43-47).
    4. Factors that influence fatigue, such as: intensity (lines 48-59), duration or time to failure (60-67) and durations of recovery within IFPs (lines 68-77).
    5. Hypothesis and Objective (lines 105-117).
  • Add manufacturer: Thank you for this comment. This information has been added into the revised manuscript (line 197-199).
  • Oxford comma; review throughout the manuscript. Thank you for this comment. We have amended this issue. (see lines 234 and 380)
  • Italic and lower case of statistics like r and p. Thank you for this comment. We have revised and amended this issue throughout the manuscript.

Reviewer 2 Report

I want to thank to the authors their effort to shed some light on one of the most important problems motorbikes riders have to face. I liked the carefully set up to test the riders and the bunch of details in which the authors have think of. Nevertheless, the paper needs major changes, specially try to rewrite the introduction, methods and results parts. I decided not to get into minors to make that revision round easier for the authors. I hope the following remarks can help them to improve their manuscript.

Introduction

  1. Line 44 and others: focus on what you want to state
  2. Line 98 and others: do not include mere details of other researches
  3. Line 76-122: state clearly that fatigue depends on intensity, duration of effort and duration of fatigue, avoid to be confusing including conflicting evidence –this can be done in the discussion chapter
  4. Hypothesis are too complicated, make them simple

Methods

  1. The level of the riders can be expressed as a number of top results, but including number of international races or hours of practice can help to define the sample
  2. Include the reference of the resolution of the Ethics Committee
  3. I liked very much the images and all the set up, but one question come to my mind, are all the elements manipulated for better customization/fitting? Whether yes or not, justify
  4. If the required 30% or 50% is adjusted or maintained visually by the subjects –as I could understand–, which is the precision of the tool (i.e. 1%) and the accepted error when performing the test –assuming that maintain a perfect score is not possible–
  5. The choice of the best MVC score as a baseline needs justification since the second score represents per se a possible state of fatigue due to its lower magnitude
  6. How the angles of the elbow and wrist have been standardized? To my knowledge that level of precision cannot be ensured through visual observation
  7. Electrodes’ placement is clearly explained, but electrode dimensions can have high impact when using EMG in certain areas
  8. In line 146, if 1’ is enough to make sure that the individuals are not affected by fatigue, the 60s rest in the protocol Is a non-sense
  9. My major concern comes from the design of the point 2.3. Did the subjects perform 6 contractions at 30% and then one MVC and then one at 50% to be back to the 30% again? If so, I cannot understand how interpret the results when both tests are carried out together
  10. Has been any randomization in the order of the tests?
  11. Although is not yet a common practice, a p-value threshold level justification is a good research practice, especially if the authors are going to draw conclusions from this value
  12. Multiple comparison methods need adjustment methods

Results

  1. For better understanding, figure 3 should have same axis scale
  2. T1 and T2 do not appear in the methods section
  3. h2 is presented but not explained and it is a good practice to include its interpretation

Author Response

Thank you to the reviewer for this constructive and encouraging feedback.

Introduction

Line 44 and others: focus on what you want to state. Thank you for this comment. We have deleted part of the sentence to relate the IFP only with the “motorcyclist forearm discomfort” (see lines 45-46). Following your suggestion, we have also strengthened the justification of the manuscript to keep focus on the main question of the study (see lines 33-37).

Line 98 and others: do not include mere details of other researches. Thank you for this comment. We have deleted “before the 10 s duration at 50% MVC” (see line 85). We have additionally deleted details from other studies (lines 73, 89).

Line 76-122: state clearly that fatigue depends on intensity, duration of effort and duration of fatigue, avoid to be confusing including conflicting evidence –this can be done in the discussion chapter. Thank you for this comment. Following your suggestions, we have substantially modified the structure of the introduction following this order:

  1. Justification of the study and association of repetitive handgrip action with functional forearm disorders (lines 24-37).
  2. Utility of focusing on IFPS because they are “common in the every-day life activities” (lines 41-43).
  3. Discrepancies between continuous and intermittent fatiguing protocols (CFP & IFP respectively; lines 43-47).
  4. Factors that influence fatigue, such as: intensity (lines 48-59), duration or time to failure (60-67) and durations of recovery within IFPs (lines 68-77).
  5. Hypothesis and Objective (lines 105-117).

Hypothesis are too complicated, make them simple. Thank you for this comment. We have simplified the main hypothesis of the manuscript: “verify that in IFPs the duration of the recovery is more relevant than the intensity and duration of the muscle contraction.” (see lines 105-107).

Methods

The level of the riders can be expressed as a number of top results, but including number of international races or hours of practice can help to define the sample. Thank you for this comment. We have struggled to make contact with all the riders that participated in this study because of the only 10 days that we were given to amend manuscript. However, we were able to reach some of the riders and in average they began to ride a motorbike at the age of 6-8 years (pocket bikes). The less experienced (regional completion level) had a minimum of five years of racing experience (line 126).  

Include the reference of the resolution of the Ethics Committee. Thank you for this comment. We have amended this issue (see lines 127-128).

I liked very much the images and all the set up, but one question come to my mind, are all the elements manipulated for better customization/fitting? Whether yes or not, justify. Yes, we did it. As it happens with a real road race motorcycle, levers, tilt, distance between levers and handle gas, distance between handle bar and seat, etc. were modified according to the ergonomic requirements of the rider (see lines 179-182).

If the required 30% or 50% is adjusted or maintained visually by the subjects –as I could understand–, which is the precision of the tool (i.e. 1%) and the accepted error when performing the test –assuming that maintain a perfect score is not possible–  As we commented in the manuscript the force level is maintained visually by the subject (lines 137-139). Regarding the precision of the tool, the linearity and hysteresis was 0.2% with a sensibility of 0.1 N (line 199).

The choice of the best MVC score as a baseline needs justification since the second score represents per se a possible state of fatigue due to its lower magnitude. Thank you for this comment. From our own experience, in this type of measurements (Marina, Rios, Torrado, Busquets, & Angulo-Barroso, 2015; Marina, Torrado, Busquets, Ríos, & Angulo-Barroso, 2013; Torrado, Cabib, Morales, Valls-Sole, & Marina, 2015), the second power grip MVC is not affected by a possible state of fatigue. This has also been shown in previous studies (Kamimura & Ikuta, 2001; Kleine, Schumann, Stegeman, & Scholle, 2000). We never had to ask for a third attempt, due to of excessive difference between both trials (≥10%). Please, see lines 141-147 for justification.

How the angles of the elbow and wrist have been standardized? To my knowledge that level of precision cannot be ensured through visual observation. We agree with the reviewer. A general description of the rider position on the motorcycle setup is provided in lines 183-189. The researcher that was responsible for the “straight layout rider position” (lines 192-193) is also an experimented rider. Standardization of this position was proposed after personal consultation with some of these “winner” professional riders. However, we recognize that that the only way of controlling rigorously the position is by using 3D system such as Qualisys, Vicon, BTS, etc…

Electrodes’ placement is clearly explained, but electrode dimensions can have high impact when using EMG in certain areas.  We agree with the reviewer. To avoid this, we carefully cut one side of the electrodes. We attach a photo for better clarification.

In line 146, if 1’ is enough to make sure that the individuals are not affected by fatigue, the 60s rest in the protocol Is a non-sense. Thank you for this comment. MVCs at baseline condition had a duration of 3-s with 5 min of resting before the beginning of the IFP, whereas the 50%MVCs had a duration of 10-s immediately followed by a resting period of 5-s that separate the actual round with the next one. We think that it is crucial for the reviewer having an historical perspective of our line of research to understand this decision. The isolated sequence of 60-s of resting + 10-s at 50%MVC was first used during a 24-h endurance race (Marina, Porta, Vallejo, & Angulo, 2011). As a consequence of this study, we designed our first IFP in laboratory with the prerequisite of maintaining this sequence within the structure of this IFP (Marina et al., 2013). We wanted to compare the data in different conditions (field and lab-based studies) so we decided to keep the same protocol to achieve this. However, it is worth to mention the knowledge acquired in previous studies is letting us to introduce improvements step by step.

My major concern comes from the design of the point 2.3. Did the subjects perform 6 contractions at 30% and then one MVC and then one at 50% to be back to the 30% again? If so, I cannot understand how interpret the results when both tests are carried out together. Following our previous answer, the only way of studying the time-course of decrement of the MVC throughout an IFP was inserting one MVC on each round. We did not want to limit the study to a simple pre and post MVC comparison [Fig. 3 of (Marina et al., 2013)]. Using to this approach, we realized afterwards that by performing a mathematical model of the force fatigue pattern with occurrence of fatigue it would allow us to categories riders according to their pattern of fatigue and to enhance their physical training (Marina et al., 2015). According to our own research experience in this field, we believe that shorter and lower %MVCs may have a higher fatiguing effect than longer and higher %MVCs. To investigate this, instead of using different types of IFPs, we decided to take the same IFP and subjects as their own control. Nevertheless, this issue appears now as a potential limitation of the study in lines 440-444.

Has been any randomization in the order of the tests? Sorry, we do not understand the question because only one test “IFP” was carried out (Fig. 1).

Although is not yet a common practice, a p-value threshold level justification is a good research practice, especially if the authors are going to draw conclusions from this value. Multiple comparison methods need adjustment methods.

We agree with the reviewer and we have expanded the statistics section for a better understanding of what has been done to arrive at the statistical results presented in Table 3 (lines 244-251).

Results

  • For better understanding, figure 3 should have same axis scale. Thank you for this comment. This issue has been amended.
  • T1 and T2 do not appear in the methods section This has been amended as well (see line 240)
  • h2 is presented but not explained and it is a good practice to include its interpretation. I presume the reviewer meant to eta squared (η2). (See lines 248-249).

Reviewer 3 Report

Overall, this study is written very well. The focus is good and I believe it will be of interest to a lot of readers. Authors articulate their rationale, findings, and explanations for findings systematically which is much appreciated given the specificity/complexity of some of the content. I only have minor recommendations for edits:

There are a lot of abbreviations. Authors should go through and make sure that all of them are necessary. Some things were abbreviated and only used once which is not needed.

Line 60: change “proved” to “but was also shown”.

Throughout the manuscript, I suggest authors change “EMG” to “sEMG” if they are indicating surface EMG.

There should be consistency when denoting “surface” and “superficial”. Assuming authors are indicating these are the same.

Line 83: The 1970 inserted in the sentence does not follow the citations guidelines (to my knowledge anyway). Authors can instead say “In the 1970’s, Mundale ….”

Line 96: Same here with the citation. Fix throughout the whole manuscript

The introduction is written quite nicely. I think the rationale is laid out very well.

Line 126: make the “k” in kg lowercase

Line 127: Do authors have years of racing experience?

Line 169: Perhaps change “top and bottom” to “a and b”?

Line 180: how was joint angle measured?

Line 218: change to a and b as mentioned above

Line 221:L authors need to state how data are displayed (i.e. mean and SD) and what the significance cutoff (alpha) was set at.

Line 331: change the word “reported” as this makes it sound like riders subjectively indicated something.

Authors should acknowledge limitations (and strengths if desired) of the study at the end of the discussion.

Author Response

Thank you to the reviewer for this constructive and encouraging feedback.

There are a lot of abbreviations. Authors should go through and make sure that all of them are necessary. Some things were abbreviated and only used once which is not needed. We recognized that we used a lot of abbreviations and particularly with the expression related to the median frequency. It was messy. Thus, and following your suggestions, we have amended this issue as follows:

    • Abbreviation FFT (Fast Fourier Transform) was deleted.
    • Abbreviations MF (Median Frequency) and NMF (Normalized median Frequency) were deleted and replaced by the only abbreviation MFEMG. For statistical analysis all data was normalized (lines 233).
    • Regarding, the EMG frequency calculated from the power spectrum density, we unified the abbreviations and finally decided to use MFEMG. MFEMG appears now 74 times. We think that the reading of the revised manuscript is more friendly.
    • Oc1 and Oc2 were a redundancy of T1 (beginning) and T2 (ending of the fatiguing protocol). Both were deleted.
    • We have deleted and simplified the abbreviation issue as much as possible. (Lines 316-338).
    • MPF, CMAP, have been also deleted.

Line 60: change “proved” to “but was also shown”. We have substantially changed the introduction following a reviewer’s suggestion. The previous message has been deleted, but we replaced “also proved to have” by “showed” (see line 46).

Throughout the manuscript, I suggest authors change “EMG” to “sEMG” if they are indicating surface EMG. This amendment has been done. sEMG appears in 16 occasions.

There should be consistency when denoting “surface” and “superficial”. Assuming authors are indicating these are the same. We have amended this issue and the word “superficial” has been replaced by sEMG.

Line 73: The 1970 inserted in the sentence does not follow the citations guidelines (to my knowledge anyway). Authors can instead say “In the 1970’s, Mundale ….”. Thank you for this comment. After the requested amendments in the introduction, Mundale’s reference appears twice into the revised manuscript using the automated Endnote format with the specific style MDPI requested by the editorial.

Line 96: Same here with the citation. Fix throughout the whole manuscript. Thank you very much for your observation. We fixed this formal aspect in line 44, 78 (Marina et al.), lines 87, 88, 418, 423 (Krogh-Lund et al.), line 398 (Nagata et al.), line 411 (Mills) and line 426 (Elfving et al.) of the revised manuscript.

The introduction is written quite nicely. I think the rationale is laid out very well. We frankly appreciate the comment of the reviewer.

Line 126: make the “k” in kg lowercase. Thank you for this comment. This issue has been amended. (Line 122).

Line 127: Do authors have years of racing experience? I presume the reviewer meant years of experience of the motorcycle rider. Reaching this level of performance (lines 123-124) they began to race in motorcycling between the age of 6-8 years. Most of them started using “pocket-bikes” (Example: https://www.youtube.com/watch?v=-dPAYxEGBk8). Anyway, following the reviewer’s suggestion, we have clarified that riders with the lowest level (regional completion) had a minimum of 5 years of racing experience (lines 126).

Line 169: Perhaps change “top and bottom” to “a and b”? A & B appears in figure 1.

Line 180: how was joint angle measured? During the pilot study a mechanical goniometer was used for the elbow joint while the more performant riders who volunteered in the study were doing partial essays of the IFP. After consistently measuring a range of 150-160º as the “natural and spontaneous position”, one of the experimenters continuously checked visually the maintenance of the body position” (lines 190-193)

Line 218: change to a and b as mentioned above. Thank you for this comment. This issue has been amended (lines 151, 228, 230).

Line 221: authors need to state how data are displayed (i.e. mean and SD) and what the significance cutoff (alpha) was set at. Thank you for this comment. This issue has been amended (lines 235,250-251).

Line 331: change the word “reported” as this makes it sound like riders subjectively indicated something. I presume the reviewer meant to “ ….. if muscle soreness is reported by the subjects” ?? Nevertheless, revising the sentence, …., structural damage is frequently diagnosed WHEN the subject reported muscle soreness (always subjective). Anyway, we appreciate this comment and we have replaced “if” by “when” (see lines 353).

Authors should acknowledge limitations (and strengths if desired) of the study at the end of the discussion. Thank you for this comment. This issue has been amended (lines 440-444).

REFERENCES USED IN THE RESPONSES TO REVIEWERS

  1. Kamimura, T., & Ikuta, Y. (2001). Evaluation of grip strength with a sustained maximal isometric contraction for 6 and 10 seconds. Journal of rehabilitation medicine : official journal of the UEMS European Board of Physical and Rehabilitation Medicine, 33(5), 225-229.
  2. Kleine, B. U., Schumann, N. P., Stegeman, D. F., & Scholle, H. C. (2000). Surface EMG mapping of the human trapezius muscle: the topography of monopolar and bipolar surface EMG amplitude and spectrum parameters at varied forces and in fatigue. Clinical neurophysiology : official journal of the International Federation of Clinical Neurophysiology, 111(4), 686-693.
  3. Marina, M., Porta, J., Vallejo, L., & Angulo, R. (2011). Monitoring hand flexor fatigue in a 24-h motorcycle endurance race. Journal of Electromyography and Kinesiology, 21(2), 255-261. doi:10.1016/j.jelekin.2010.11.008
  4. Marina, M., Rios, M., Torrado, P., Busquets, A., & Angulo-Barroso, R. (2015). Force-time course parameters and force fatigue model during an intermittent fatigue protocol in motorcycle race riders. Scandinavian journal of medicine & science in sports, 25(3), 406-416. doi:10.1111/sms.12220
  5. Marina, M., Torrado, P., Busquets, A., Ríos, J. G., & Angulo-Barroso, R. (2013). Comparison of an intermittent and continuous forearm muscles fatigue protocol with motorcycle riders and control group. Journal of Electromyography and Kinesiology, 23(1), 84-93. doi:10.1016/j.jelekin.2012.08.008
  6. Torrado, P., Cabib, C., Morales, M., Valls-Sole, J., & Marina, M. (2015). Neuromuscular Fatigue after Submaximal Intermittent Contractions in Motorcycle Riders. International journal of sports medicine, 36(11), 922-928. doi:10.1055/s-0035-1549959

Round 2

Reviewer 2 Report

I want to congratulate the authors since this new version clearly improved the original one. Nevertheless, I still see some issues than, in my opinion, need to be addressed:

  • The fact that CR-50 increases with the rounds, contrary to the expected apparition of fatigue –actually looks more like a potentiation effect–, is probably, the strongest point to defend the use of an intermittent protocol over the continuous ones; in my opinion authors miss to highlight this finding, I want to encourage them to emphasize it as a major finding (stronger correlations, in my opinion are not that relevant)
  • In the abstract and discussion i.e. lines 11-12 is stated that fatigue occurs at 30% but not at 50%, to be fair intermittent and continuous should be specified to avoid understand that a lower MVC% elicits fatigue over a higher percentage
  • I was not able to find where the authors discuss the different effect sizes, that –again–, can help to strengthen their conclusions/findings
  • Lines 296-307, biochemical explanation of fatigue is not relevant and it is not appropriate at the beginning of the discussion because it has nothing to do with the results reported in the work; this work’s findings are fine enough to compose a nice discussion
  • L 318, relationships are stated firmly through repetitive research results across multiple studies, what you found in the study could be a ‘stronger correlation’
  • When reporting p-values use the less than symbol below a specified threshold and the equal to or no symbol when reporting exact values
  • Minor grammar, English and format corrections are also necessary

I hope the present remarks can help the authors to finish their manuscript.

Author Response

  • The fact that CR-50 increases with the rounds, contrary to the expected apparition of fatigue –actually looks more like a potentiation effect–, is probably, the strongest point to defend the use of an intermittent protocol over the continuous ones; in my opinion authors miss to highlight this finding, I want to encourage them to emphasize it as a major finding (stronger correlations, in my opinion are not that relevant).

We honestly appreciate the comments of the reviewer concerning the major finding. His comments denote that he has caught the idea that we want to broadcast. In our opinion, the tricky point is that without using different protocols, within the same IFP for the same subject during the same assessment of the same day, we noticed that in some sections (of more intermittent nature: 5 s of work / 5 s of rest at 30% = duty cycle of 100%) the participants were getting more tired than in other sections (more sustained: 60 s of rest / 10 s work at 50% = duty cycle of 16%) (see lines 19-21). We added the adjective “comparative” ANOVA analysis to emphasize that we carried out two approaches for the same objective (lines: 12-13). Regression analysis, and 2) comparison between times of measurement (beginning -T1- and ending -T2-) (Table 3). We hope that the second part of the abstract (lines 10-19) reinforce better this idea.

  • In the abstract and discussion i.e. lines 11-12 is stated that fatigue occurs at 30% but not at 50%, to be fair intermittent and continuous should be specified to avoid understand that a lower MVC% elicits fatigue over a higher percentage.

It is clear that we did not precise sufficiently the message. We propose adding the duration of each intensity to highlight the intermittent nature of the 30%MVC section in comparison to the 50%MVC brief (10 -s) continuous one. In lines 19-21 we emphasized this different nature of intermittency versus continuity.   

  • I was not able to find where the authors discuss the different effect sizes, that –again–, can help to strengthen their conclusions/findings

It is true that we report values of effect size (lines 276-277) and Table 3, but we did not discuss or propose interpretation of them in the result section. Done (see lines 345, 364-366). Because SPSS software report Partial eta square (line 276), we also decided to replace η2 by η2p.

  • Lines 296-307, biochemical explanation of fatigue is not relevant and it is not appropriate at the beginning of the discussion because it has nothing to do with the results reported in the work; this work’s findings are fine enough to compose a nice discussion

Following your suggestions, we have deleted the biochemical information (see lines 378-381)

  • L 318, relationships are stated firmly through repetitive research results across multiple studies, what you found in the study could be a ‘stronger correlation’

Done (see line 400-401)

  • When reporting p-values use the less thansymbol below a specified threshold and the equal to or no symbol when reporting exact values

Done. We have modified throughout the text, particularly in the result section.

  • Minor grammar, English and format corrections are also necessary.

We recognize that some mismatch and grammatical errors were present in the previous version. We have thoroughly revised the text with an English expert.

  • I hope the present remarks can help the authors to finish their manuscript. Absolutely yes. We firmly believe that after two revisions the original manuscript has been substantially improved. Thank you for your time and expertise.